# LOCALIZING AND AMORTIZING:
# EFFICIENT INFERENCE FOR GAUSSIAN PROCESSES

## ABSTRACT

The inference of Gaussian Processes concerns the distribution of the underlying function given observed data points. GP inference based on local ranges of data points is able to capture fine-scale correlations and allow fine-grained decomposition of the computation. Following this direction, we propose a new inference model that considers the correlations and observations of the $K$ nearest neighbors for the inference at a data point. Compared with previous works, we also eliminate the data ordering prerequisite to simplify the inference process. Additionally, the inference task is decomposed to small subtasks with several technique innovations, making our model well suits the stochastic optimization. Since the decomposed small subtasks have the same structure, we further speed up the inference procedure with amortized inference. Our model runs efficiently and achieves good performances on several benchmark tasks.

## 1 INTRODUCTION

Gaussian processes (GP) (Rasmussen & Williams, 2006) are flexible non-parametric models with a wide range of applications. GP poses a Gaussian prior over function values $\mathbf{f}$ and assumes observations $\mathbf{y}$ are generated independently given $\mathbf{f}$. GP inference considers the calculation of the posterior of these function values (Matthews et al., 2016) given observations, namely $p(\mathbf{f}|\mathbf{y})$. Direct computation of the posterior is often intractable on large datasets, motivating people to consider its approximations. Variational inference (Jordan et al., 1999; Blei et al., 2017) for GP (Rasmussen & Williams, 2006) has achieved great successes recently. Variational inference constructs a *variational distribution*, which is usually a multivariate Gaussian distribution, to approximate the posterior. The approximation is done by minimizing the KL divergence from the posterior to the variational distribution (Blei et al., 2017). The variational distribution is often constructed with some special structures to reduce the number of variational parameters and speed up the computation.

Inducing-point methods (Quiñonero-Candela & Rasmussen, 2005; Titsias, 2009; Hensman et al., 2013; 2015) define variational distributions on a small number $M$ of *inducing points* and then derive the distribution of non-inducing points conditioned on these inducing points. Inducing points summarize the entire posterior distribution, and their number $M$ balances the computational cost and the quality of the approximation. Inducing-point methods are further improved in several directions, such as generic inference for non-Gaussian likelihoods (Sheth et al., 2015; Dezfouli & Bonilla, 2015; Krauth et al., 2016; Hensman et al., 2015), inter-domain and subspace inducing points (Hensman et al., 2017; Panos et al., 2018), and decoupled approximation with two different sets of inducing points (Cheng & Boots, 2017; Salimbeni et al., 2018). Burt et al. (2019) provide theoretical analysis to show that a relatively small $M$ is sufficient to produce a reliable variational approximation when the input dimension is low.

While inducing-point methods capture global correlations among data points through inducing points, inference methods based on local neighbors focus more on correlation structures at local scales. These methods consider only local-range dependencies to save computation because local-range correlations are often much stronger than distant ones. Nguyen-Tuong et al. (2009); Park & Apley (2018) partition the input space into subregions, fit local models over subregions and then stitch local models into one. Other works examine neighbors of each data point directly. Gramacy & Apley (2015) investigate the properties of GP predictive equation and construct a local predictive approximator. Covariance tapering (Furrer et al., 2006; Kaufman et al., 2008) gains computational

efficiency by constructing a sparse correlation matrix with zero correlations between distant data points. Methods based on Vecchia's approximation (Vecchia, 1988; Datta et al., 2016; Liu & Liu, 2019; Finley et al., 2019) decompose the joint probability of data points into conditionals according to a data ordering and then neglect far data points that are conditioned on.

Recently, Liu & Liu (2019) propose the AIGP method, which extends the idea of local inference to GP models with non-Gaussian likelihoods. They use directed graphical models to approximate both the prior and the posterior. With this construction, the inference task decomposes into local inference subtasks, then they introduce amortized inference and use inference networks to identify solutions to these subtasks (Kingma & Welling, 2013; Dai et al., 2015; Miao et al., 2016). Amortization reduces the number of optimization parameters and greatly speeds up the inference procedure. However, this method has two drawbacks. First, the inference at a data point considers a few of its nearest neighbors but not all of them; therefore, it may lose some important correlations. Second, it depends on a data ordering. A bad ordering often deteriorates the performance, but it is hard to guard against such a bad situation. There are no easy fixes of the two issues, because all these designs in AIGP serve the purpose of decomposition.

In this work, we propose a new GP inference method, Localized and Amortized Inference based on Nearest neighbors (LAIN). LAIN considers $K$ nearest neighbors for the inference at each data point. Particularly, LAIN uses a variational distribution whose covariance is parameterized by a sparse decomposition. The decomposition focuses on the correlations between every data point and its $K$ nearest neighbors [1]. LAIN also eliminates the need for a data ordering. These nice properties come after several technical innovations. First, the new distribution does not admit a decomposable entropy calculation. We overcome this difficulty by using a decomposable lower bound of the entropy (Ranganath et al., 2016; Louizos & Welling, 2017). Second, to decompose the logarithm of the prior, AIGP and previous methods use a directed graphical model as an approximation of the prior. We follow this idea, but we consider all possible orderings of data points and collapse them to local combinations, making the computation manageable. With these techniques, LAIN still decomposes the inference task into subtasks, so amortized inference can apply. It is worthing noting that subtasks in LAIN are generated from the same mechanism while those in AIGP are not. We argue that subtasks sharing the same "distribution" are more appropriate for amortization.

Our empirical evaluations show that the LAIN method outperforms baseline methods including AIGP in several learning tasks. Our investigation also indicates that LAIN can achieve decent performance even only a few neighbors are considered.

## 2 BACKGROUND

**Gaussian Processes.** Suppose we have a dataset containing a feature matrix $\mathbf{X} = (\mathbf{x}_i)_{i=1}^N$ and observations $\mathbf{y} = (y_i)_{i=1}^N$. We assume there is a latent function $f$ that generates $y_i$ from $\mathbf{x}_i$ for each $i$. Particularly, each $y_i$ is generated by a likelihood model $p(y_i|f_i)$ with $f_i = f(\mathbf{x}_i)$. Denote $\mathbf{f} = (f_i)_{i=1}^N$, then $p(\mathbf{y}|\mathbf{f}) = \prod_{i=1}^N p(y_i|f_i)$. The likelihood $p(y_i|f_i)$ can be very general – here we only assume that $\log p(y_i|f_i)$ is differentiable with respect to $f_i$. This mild assumption allows a wide range of data distributions. For example, if $y_i$ is binary, $p(y_i|f_i)$ is a Bernoulli distribution with $f_i$ as the logit.

We put a GP prior with a mean function $\nu(\cdot)$ and a kernel function $\kappa(\cdot, \cdot)$ over the latent function $f$. The kernel function encodes the prior knowledge of the smoothness of $f$. One commonly used kernel function is the Radial Basis Function (RBF) kernel, $\kappa(\mathbf{x}_i, \mathbf{x}_j) = r^2 \exp(-0.5\|\mathbf{x}_i - \mathbf{x}_j\|_2^2/\sigma^2)$, with $r$ and $\sigma$ as parameters. With this prior, function values in $\mathbf{f}$ follow a multivariate Gaussian, $\mathbf{f} \sim \mathcal{N}(\boldsymbol{\nu}, \boldsymbol{\Sigma})$, with the mean $\boldsymbol{\nu} = (\nu(x_i))_{i=1}^N$ and the covariance matrix $\boldsymbol{\Sigma}$ with $\Sigma_{i,j} = \kappa(\mathbf{x}_i, \mathbf{x}_j) \, \forall i, j$.

GP inference concerns the calculation of the posterior $p(\mathbf{f}|\mathbf{y})$ (Matthews et al., 2016), from which we can infer the function value $f_\star$ for any new input $\mathbf{x}_\star$ with integral $\int_{\mathbf{f}} p(f_\star|\mathbf{f})p(\mathbf{f}|\mathbf{y})\mathrm{d}\mathbf{f}$. The posterior $p(\mathbf{f}|\mathbf{y})$ is generally not tractable, so we appeal to approximate inference.

**Variational Inference for GP.** Variational inference approximates the posterior $p(\mathbf{f}|\mathbf{y})$ with a variational distribution $q(\mathbf{f})$, which is defined as a multivariate Gaussian distribution, $q(\mathbf{f}) \sim \mathcal{N}(\boldsymbol{\mu}, \mathbf{V})$.

---

[1]We use the term "nearest neighbors" for easy reference, but we actually consider $K$ most correlated neighbors in the prior. When the kernel is based on some distance metrics, then they are $K$ nearest neighbors.

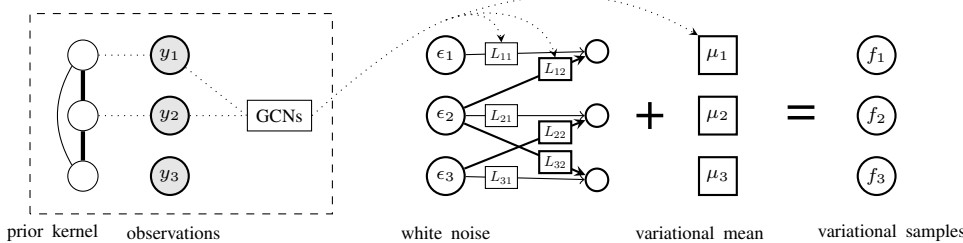

prior kernel   observations          white noise          variational mean      variational samples

**Figure 1:** The structure of the variational distribution. The left box shows the amortization, which fits $\mu_i$-s and $R_{ij}$-s from their related prior kernel and observations. The right part shows the generation process of $f_i$-s.

The inference is carried out by maximizing the Evidence Lower BOund (ELBO) with respect to $q(\mathbf{f})$ (Blei et al., 2017).

$$\log p(\mathbf{y}|\mathbf{X}) \geq \max_{q(\mathbf{f})} \underbrace{\mathbb{E}_q\left[\log p(\mathbf{y}|\mathbf{f})\right]}_{L_{ell}} + \underbrace{\mathbb{E}_q\left[\log p(\mathbf{f})\right]}_{L_{cross}} \underbrace{-\mathbb{E}_q\left[\log q(\mathbf{f})\right]}_{L_{ent}} \tag{1}$$

Here we name the three terms in the ELBO for easy reference later. Typically the ELBO is maximized by gradient-based optimization, preferably stochastic gradient optimization when $N$ is large. Direct optimization of the ELBO is challenging, since the kernel matrix $\boldsymbol{\Sigma}$ and the variational covariance $\mathbf{V}$ are both large and have size $N \times N$.

Inducing-point methods define $q(\mathbf{f}) = \int_{\mathbf{f}_I} q(\mathbf{f}_I)p(\mathbf{f}|\mathbf{f}_I)\,\mathrm{d}\mathbf{f}_I$, where $q(\mathbf{f}_I)$ is the distribution over inducing points $I$, and $p(\mathbf{f}|\mathbf{f}_I)$ is derived from the prior. The computation is reduced mainly because only the small distribution $q(\mathbf{f}_I)$ is optimized, while the conditional $p(\mathbf{f}|\mathbf{f}_I)$ is fixed when the prior is given.

AIGP parameterizes $\mathbf{V}$ by a Cholesky decomposition, $\mathbf{V} = \mathbf{L}\mathbf{L}^\top$. Here $\mathbf{L}$ is a sparse lower triangular matrix, and each row of $\mathbf{L}$ has at most $K$ non-zero entries. AIGP uses a triangular $\mathbf{L}$ for easy entropy computation. It also approximates $\log p(\mathbf{f})$ with a directed graphical model. Both the lower triangular matrix $\mathbf{L}$ and the directed graph require an ordering of data points.

## 3   METHOD

### 3.1   THE VARIATIONAL DISTRIBUTION

Following previous works, we also define the variational distribution $q(\mathbf{f})$ to be a multivariate Gaussian $\mathcal{N}(\boldsymbol{\mu}, \mathbf{V})$. We parameterize $\mathbf{V} = \mathbf{R}\mathbf{R}^\top + \delta^2 \mathbf{I}$ with $\mathbf{R}$ being a sparse matrix and $\delta$ being a small constant. Note that we do not require $\mathbf{R}$ to be triangular. The sparse pattern of $\mathbf{R}$ is decided by the nearest neighbors: $R_{ij} \neq 0$ only when $j \in n(i)$. Here $n(i)$ is the neighbor set containing data points that have the largest covariance with $i$ in the prior (by definition $n(i)$ includes $i$). In this work, we fix the size of $n(i)$ to be $K$, though our derivation works for varied sizes of $n(i)$. The row $\mathbf{R}_i$ can be viewed as a representation of $f_i$ in the variational distribution: $\mathbf{R}_i$ informs $f_i$'s correlation with other function values, just like a word embedding informs its relation with other words (Mikolov et al., 2013).

Efficient sampling from the marginal is critical for the decomposition of the ELBO later. Owing to the sparse decomposition of the covariance matrix, we can cheaply draw marginal samples for an $f_i$ from $q(\mathbf{f})$ with a linear transformation of white noise. The sampling scheme is shown in (2) and pictured in the right part of Figure 1.

$$f_i = \mu_i + \mathbf{R}_i\,\boldsymbol{\epsilon} + \delta\xi = \mu_i + \mathbf{R}_{i,n(i)}\,\boldsymbol{\epsilon}_{n(i)} + \delta\xi, \ \ \boldsymbol{\epsilon} \sim \mathcal{N}(\mathbf{0}, \mathbf{I}), \ \ \xi \sim \mathcal{N}(0, 1). \tag{2}$$

The constructed distribution $q(\mathbf{f})$ well approximates the strong correlations in the prior. From (2), $f_i$ and $f_j$ correlate in $q(\mathbf{f})$ by sharing noise entries in $n(i) \cap n(j)$ when the intersection is not empty. In this case, either $f_i$ neighbors $f_j$, or $f_j$ neighbors $f_i$, or $f_i, f_j$ share common neighbors. When the neighbor sets are large enough, most strong correlations will be approximated by some non-zero entries in $\mathbf{V}$.

### 3.2 OPTIMIZATION OF THE ELBO

We optimize the ELBO in (1) to find a good $q(\mathbf{f})$ to approximate the GP posterior. To apply stochastic optimization, we will decompose the three terms in the ELBO. We mainly consider the decomposition of $L_{cross}$ and $L_{ent}$, as the decomposition of $L_{ell}$ is easy.

We first decompose the cross entropy $L_{cross}$. By convention, the GP prior has a zero mean. Though there is a closed-form calculation of $L_{cross}$ with both $q(\mathbf{f})$ and $p(\mathbf{f})$ being multivariate Gaussian, it involves expensive calculations of $\det(\boldsymbol{\Sigma})$ and $\boldsymbol{\Sigma}^{-1}$. Previous works approximate the prior with Vecchia's method for easy decomposition and good approximation (Vecchia, 1988; Stein et al., 2004; Datta et al., 2016; Liu & Liu, 2019; Finley et al., 2019). The idea is to build a directed graphical model and approximate $p(\mathbf{f}) \approx \prod_{i=1}^{N} p(f_i | f_{\alpha(i)})$ with $\alpha(i)$ being a small parent set of $i$. In the original work, Vecchia (1988) first set an order to data points and then choose $\alpha(i)$ as the $K$ nearest parents of $i$. But it is not easy to guarantee a good ordering of data points (Banerjee et al., 2014; Guinness, 2018). Here we consider all possible orderings and take the average of approximations to address the data ordering concern.

We estimate $L_{cross}$ as follows. First, we randomly sample a *parent set* $n'(i) \subset n(i)$ with $i \notin n'(i)$ for each $i$. Then we approximate the log-prior by $\log p(\mathbf{f}) \approx \sum_{i=1}^{N} \log p(f_i | f_{n'(i)})$, with the conditional distribution $p(f_i | f_{n'(i)})$ derived from the joint Gaussian $p(f_i, f_{n'(i)})$ in the prior. Then $L_{cross}$ is estimated by a random batch of terms. The complete calculation is given as

$$L_{cross} \approx \tilde{L}_{cross} = \frac{N}{|S|} \sum_{i \in S} \mathbb{E}_{q(f_i, f_{n'(i)})} \Big[ \log p(f_i | f_{n'(i)}) \Big], \quad \text{random set } n'(i) \subset n(i). \quad (3)$$

Here $S$ is a random batch of data points.

Now we justify that this is an average over all data orderings. Suppose there is a data order $\pi(\cdot)$, such that we can define a directed graphical model over $p(\mathbf{f})$ by assigning every $i$ a parent set $n'_\pi(i) = \{j : j \in n(i), \pi(j) < \pi(i)\}$. Denote $\Pi$ as all permutations of $N$ data points, with each permutation inducing a graphical model. The average of the log densities of all graphical models can be collapsed to the average computed from local neighborhoods. Denote $\Pi_{n(i)}$ as permutations of indices in the set $n(i)$, then we have

$$\frac{1}{N!} \sum_{\pi \in \Pi} \sum_{i=1}^{N} \mathbb{E}_{q(f_i, f_{n'_\pi(i)})} \Big[ \log p(f_i | f_{n'_\pi(i)}) \Big] = \sum_{i=1}^{N} \frac{1}{K!} \sum_{\pi \in \Pi_{n(i)}} \mathbb{E}_{q(f_i, f_{n'_\pi(i)})} \Big[ \log p(f_i | f_{n'_\pi(i)}) \Big]. \quad (4)$$

Here we only need to consider permutations of data points within $n(i)$ for each $i$. Then we obtain (3) by estimating the inner summation by a single random permutation of $n(i)$ and the outer summation by a random batch $S$.

We then decompose the entropy $L_{ent}$. The entropy of $q(\mathbf{f})$ requires the expensive computation of $\det(\mathbf{V})$. To circumvent this difficulty, we find a decomposable lower bound of the entropy by using an auxiliary distribution (Ranganath et al., 2016; Louizos & Welling, 2017). Note that we always prefer a lower bound of the objective in this maximization problem. With an arbitrary distribution $r(\boldsymbol{\epsilon}|\mathbf{f})$, a lower bound of $L_{ent}$ is

$$L_{ent} = -\mathbb{E}_q \left[ \log q(\mathbf{f}) \right] \geq -\mathbb{E}_{q(\mathbf{f}, \boldsymbol{\epsilon})} \left[ \log q(\mathbf{f}|\boldsymbol{\epsilon}) + \log q(\boldsymbol{\epsilon}) - \log r(\boldsymbol{\epsilon}|\mathbf{f}) \right]. \quad (5)$$

The bound is tight when $r(\boldsymbol{\epsilon}|\mathbf{f})$ matches $q(\boldsymbol{\epsilon}|\mathbf{f})$. In this work, we try to let $r(\boldsymbol{\epsilon}|\mathbf{f})$ match $q(\boldsymbol{\epsilon}|\mathbf{f})$. Particularly, we set $r(\boldsymbol{\epsilon}|\mathbf{f}) = \prod_i q(\epsilon_i | \mathbf{f}_{n(i)})$, where the conditional $q(\epsilon_i | \mathbf{f}_{n(i)})$ is derived from the joint Gaussian distribution $q(\epsilon_i, \mathbf{f}_{n(i)})$. Then all terms in the lower bound in (5) are Gaussian log-likelihoods and are decomposable over data points. We can then reach the estimation of the entropy lower bound with a batch of data points.

$$L_{ent} \geq \tilde{L}_{ent} = -\frac{1}{2} \frac{N}{|S|} \sum_{i \in S} \log \left( 1 - \mathbf{R}_{n(i),i}^\top \left( \mathbf{R}_{n(i),:} \mathbf{R}_{n(i),:}^\top \right)^{-1} \mathbf{R}_{n(i),i} \right) + const. \quad (6)$$

We finally decompose the likelihood $L_{ell}$. The likelihood term $\log p(\mathbf{y}|\mathbf{f})$ naturally decomposes because $y_i$-s are conditionally independent given $f_i$-s.

$$L_{ell} = \sum_{i=1}^{N} \mathbb{E}_{q(f_i)} \left[ \log p(y_i | f_i) \right], \quad \tilde{L}_{ell} = \frac{N}{|S|} \sum_{i \in S} \log p(y_i | \hat{f}_i). \quad (7)$$

Here for each term $i$ in the summation, the expectation is estimated by a Monte Carlo sample $\hat{f}_i$ from $q(f_i)$. The gradients of variational parameters are propagated through $\hat{f}_i$ via reparameterization (Kingma & Welling, 2013).

Finally, the ELBO has a decomposable approximation $\tilde{L}_{ell} + \tilde{L}_{cross} + \tilde{L}_{ent}$ to enable efficient stochastic optimization. From the derivations above, we see the objective can be decomposed by data points. The computation for a data point only involves itself and its $K$ nearest neighbors. Therefore, each stochastic gradient calculation takes time only $O(K^3)$. There are $N(K+1)$ parameters in $\boldsymbol{\mu}$ and $\mathbf{R}$ to optimize, so the optimization takes at least $O(N)$ time. We further reduce the number of parameters by amortizing the cost through a shared inference model, taking advantage of the fact that the inference for each data point $i$ only needs its $K$ nearest neighbors.

### 3.3 Amortized Inference

Following AIGP, we also apply amortized inference to GP inference. Particularly, we train an inference network to identify variational parameters ($\mu_i$ and $\mathbf{R}_{i,n(i)}$) for each data point $i$. Since node correlations at a neighborhood can be easily treated as a weighted graph, we use Graph Convolutional Networks (GCNs) (Kipf & Welling, 2017) as our inference network.

A GCN takes an adjacency matrix $\mathbf{A} \in \mathbb{R}^{k \times k}$ of graph and the node features $\mathbf{H}^{(0)} \in \mathbb{R}^{k \times d_0}$ as the input and then makes predictions for all graph nodes. Let $\bar{\mathbf{A}}$ be the normalized adjacency matrix, $\bar{\mathbf{A}} = \mathbf{D}^{-\frac{1}{2}} \mathbf{A} \mathbf{D}^{-\frac{1}{2}}$, with $\mathbf{D}$ being the diagonal degree matrix. A GCN layer $\ell$ with the input $\mathbf{H}^{(\ell-1)}$ is defined by $\mathbf{H}^{(\ell)} = g_\ell(\mathbf{H}^{(\ell-1)}, \mathbf{A}) := \sigma\left(\bar{\mathbf{A}} \mathbf{H}^{(\ell-1)} \mathbf{W}^{(\ell)}\right)$. Here $\mathbf{W}^{(l)} \in \mathbb{R}^{d_{\ell-1} \times d_\ell}$ is the weight matrix of the layer $\ell$. $\sigma(\cdot)$ is the activation function. An $L$-layer GCN computes its output by $\mathbf{H} = gcn(\mathbf{H}^0, \mathbf{A}) := g_L(\ldots g_1(\mathbf{H}^0, \mathbf{A}) \ldots, \mathbf{A})$. We use two GCNs for the inference task, $gcn_1$ for the calculation of $\mu_i$ and $gcn_2$ for $\mathbf{R}_{i,n(i)}$:

$$\mu_i = \mathbf{a}^\top gcn_1\left([\mathbf{y}_{n(i)}, \mathbf{e}_i], \boldsymbol{\Sigma}_{n(i),n(i)}\right), \mathbf{R}_{i,n(i)} = gcn_2\left([\mathbf{y}_{n(i)}, \mathbf{e}_i], \boldsymbol{\Sigma}_{n(i),n(i)}\right). \tag{8}$$

Here we use $\boldsymbol{\Sigma}_{n(i),n(i)}$ as the adjacency matrix and stack the observation $\mathbf{y}_{n(i)}$ and the one-hot vector $\mathbf{e}_i$ as the input feature. The vector $\mathbf{e}_i$ indicates the element $i$ for which the inference is running for. We choose the activation $\sigma(\cdot)$ to be ReLU for intermediate layers and identity for the last layer. The last layer of each GCN has size 1 to output a $K \times 1$ vector. $\mathbf{a}$ is an averaging vector with all $K$ elements as $\frac{1}{K}$. The dashed box in Figure 1 shows the amortization.

LAIN defines an inference subtask on a data point and its *nearest neighbors*, while AIGP defines a subtask on a data point and its *parents*. Due to this difference, LAIN has two advantages. First, the inference network of LAIN uses the observations from all the $K$ nearest neighbors, while the inference network of AIGP uses observations from parents only but not children. Second, inference subtasks of LAIN are generated with the same mechanism because the nearest-neighbor relationship is homogeneous across all data points. However, the parent-child relationship in AIGP depends on the ordering of data points (e.g. the first one in the order does not have parents). As a learning model, the inference network prefers subtasks from the same "distribution".

The computational cost of GCN is $O(K^2)$ by treating the network size as constant. The complexity of one gradient calculation is $O(K^3)$. The optimization procedure converges fast since it only optimizes a constant number of variational parameters. In practice, we often observe that the optimization procedure converges in less than one epoch, which is not possible for methods without amortization. Finding nearest neighbors is the only step with running time bounds to the data size, but it only needs one run and is often fast on medium to large data sizes. If the data has a very large size, we can use k-d trees for low-dimensional data and approximate algorithms (Arya et al., 1998; Datar et al., 2004) for high dimensional data.

### 3.4 Prediction

For a new data point $\mathbf{x}_\star$ with its $K$ nearest neighbors $n(\star)$ in the prior, the predictive distribution is

$$p(y_\star | \mathbf{x}_\star, \mathbf{X}, \mathbf{y}) \approx \int_{f_\star} p(y_\star | f_\star) q(f_\star | \mathbf{x}_\star, \mathbf{X}_{n(\star)}, \mathbf{y}_{n(\star)}) \mathrm{d} f_\star \approx \frac{1}{|F|} \sum_{\hat{f}_\star \in F} p(y_\star | \hat{f}_\star). \tag{9}$$

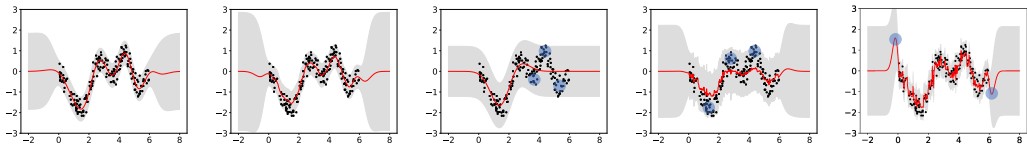

**Figure 2:** The first two plots compare predictive distributions of full GP and LAIN with $K = 10$. The right three plots show how SVGP, AIGP, and LAIN perform with a very small number of inducing points/neighbors. Data points in blue circles are not well fitted.

Here $q(f_\star | \mathbf{x}_\star, \mathbf{X}_{n(\star)}, \mathbf{y}_{n(\star)}) = \int_{\mathbf{f}_{n(\star)}} p(f_\star | \mathbf{f}_{n(\star)}) q(\mathbf{f}_{n(\star)}) \mathrm{d} \mathbf{f}_{n(\star)}$ is a Gaussian with parameters,

$$\mu_\star = \mathbf{b}_\star \mu_{n(\star)}, \qquad \sigma_\star^2 = \boldsymbol{\Sigma}_{\star,\star} - \boldsymbol{\Sigma}_{\star, n(\star)} \mathbf{b}_\star^\top + \mathbf{b}_\star (\mathbf{R}_{n(\star)} \mathbf{R}_{n(\star)}^T) \mathbf{b}_\star^\top, \tag{10}$$

with $\mathbf{b}_\star = \boldsymbol{\Sigma}_{\star, n(\star)} \boldsymbol{\Sigma}_{n(\star), n(\star)}^{-1}$. $F$ is a set of Monte Carlo samples from $q(f_\star | \mathbf{x}_\star, \mathbf{X}_{n(\star)}, \mathbf{y}_{n(\star)})$. The Monte Carlo estimation is accurate since the integral is one-dimensional.

## 4 EXPERIMENT

We compare our method with five state-of-the-art methods: SVGP (Hensman et al., 2015), SAVIGP (Dezfouli & Bonilla, 2015), DGP (Cheng & Boots, 2017), VFF (Hensman et al., 2017), and AIGP (Liu & Liu, 2019). The first three methods are based on inducing points, VFF uses inter-domain inducing points, and AIGP uses local neighbors. Through all experiments, we use RBF as the default kernel, except for VFF we use Matérn-$\frac{3}{2}$ kernel (the code does not provide RBF kernel). We use the implementation of SVGP from GPFlow (Matthews et al., 2017), the implementation of DGP from Faust (2018), and implementations of all other algorithms from their authors.

For SVGP, SAVIGP, and VFF, we vary the number of inducing points, $M \in \{200, 1000, 2000\}$, to check their performances. DGP has separate inducing points for mean approximation and those for variance approximation. We use 256 inducing points for variance approximation and vary the number of inducing points for mean approximation from 200 to 2000. We vary the number of neighbors, $K \in \{10, 20, 40\}$, for AIGP and LAIN. GCNs used in these two methods have three hidden layers with dimensions [20, 10, 1]. We randomly split each dataset into training (75%) and testing (25%) and report both the predictive performance on the test set and the inference running time. To save the space, we report results from two settings for each competing method: one setting is $M = 200$ or $K = 10$, with which all methods have their fastest speed (marked by ⚡), and another setting giving the best predictive performance (marked by ✓).

### 4.1 A TOY EXAMPLE

In this section, we test different methods on a one-dimensional toy example studied in (Snelson & Ghahramani, 2006). The dataset contains 200 data points, shown as black dots in figure 2. We assume Gaussian likelihood in this experiment and run exact inference as the baseline. A smaller GCN (hidden dimensions [10, 5, 1]) is used in this task.

The predictive mean and variance from the exact inference and LAIN with $K = 10$ are shown in the first two plots of Figure 2. The result of LAIN is very similar to that of exact inference, except that the mean curve of LAIN is less smooth, which does not really hurt the predictive performance.

We test different methods with very small $M$ and $K$ and observe how they behave. We are likely to face this situation when we work on large datasets in high-dimensional spaces. The last three plots of Figure 2 exhibit predictive distributions of SVGP with $M = 2$ inducing points, AIGP with $K = 2$ parents, and LAIN with $K = 2$ nearest neighbors. When there are not enough inducing points, SVGP over-smooths the prediction and performs poorly for a good fraction of data points. AIGP does not have a good predictive mean either, because under a random ordering the directed graph constructed by AIGP cannot well capture neighboring relations. The predictive mean of LAIN does not deviate far from the ground-truth in the area with training instances, though the curve is rugged due to local variations.

**Table 1:** Comparison on the eBird dataset.

| Method | Config | | Pred NLL | Time |
|---|---|---|---|---|
| SVGP | M=200 | ⚡ | 1.90±.03 | 107s |
| | M=1000 | ✓ | 1.88±.02 | 4.5ks |
| SAVIGP | M=200 | ⚡ | 2.04±.03 | 167s |
| | M=2000 | ✓ | 1.99±.03 | 50ks |
| VFF | M=200 | ⚡ | 1.91±.02 | 1.3ks |
| | M=2000 | ✓ | 1.91±.02 | 13ks |
| DGP | M=200 | ⚡ | 1.82±.02 | 96s |
| | M=2000 | ✓ | 1.80±.02 | 213s |
| AIGP | K=10 | ⚡ | 1.79±.05 | 45s |
| | K=20 | ✓ | 1.71±.05 | 125s |
| LAIN | K=10 | | 1.69±.03 | 55s |
| | K=20 | | 1.65±.03 | 384s |
| | K=40 | | **1.60±.02** | 1.3ks |

**Table 2:** Comparison on the precipitation dataset.

| Method | Config | | Pred NLL | Time |
|---|---|---|---|---|
| SVGP | M=200 | ⚡ | 1.57±.03 | 2.5ks |
| | M=2000 | ✓ | 1.28±.03 | 42ks |
| SAVIGP | M=200 | ⚡ | 1.70±.02 | 2.8ks |
| | M=2000 | ✓ | 1.58±.02 | 50ks |
| VFF | M=200 | ⚡ | 1.54±.03 | 9.1ks |
| | M=2000 | ✓ | 1.53±.03 | 32ks |
| DGP | M=200 | ⚡ | 1.07±.05 | 402s |
| | M=2000 | ✓ | 1.00±.05 | 889s |
| AIGP | K=10 | ⚡ | 0.96±.03 | 155s |
| | K=10 | ✓ | 0.96±.03 | 155s |
| LAIN | K=10 | | 0.74±.05 | 129s |
| | K=20 | | 0.72±.05 | 903s |
| | K=40 | | **0.69±.04** | 2.3ks |

**Table 3:** Comparison on the MNIST dataset.

| Method | Config | | Pred NLL | Accuracy | Time |
|---|---|---|---|---|---|
| SVGP | M=200 | ⚡ | 0.053±.004 | 98.4 | 623s |
| | M=1000 | ✓ | 0.051±.004 | 98.5 | 23ks |
| SAVIGP | M=200 | ⚡ | 0.339±.008 | 51.7 | 6.5ks |
| | M=200 | ✓ | 0.339±.008 | 51.7 | 6.5ks |
| DGP | M=200 | ⚡ | 0.059±.005 | 98.1 | 292s |
| | M=2000 | ✓ | 0.052±.005 | 98.3 | 2.1ks |
| AIGP | K=10 | ⚡ | 0.293±.002 | 98.0 | 3.9ks |
| | K=40 | ✓ | 0.215±.003 | 98.2 | 24ks |
| LAIN | K=10 | | 0.053±.003 | 98.9 | 128s |
| | K=20 | | **0.050±.003** | 99.0 | 632s |
| | K=40 | | 0.051±.003 | **99.1** | 2.9ks |
| KNN | K=9 | | N.A. | 98.6 | 24s |
| | K=19 | | N.A. | 98.3 | 26s |
| | K=39 | | N.A. | 97.6 | 28s |

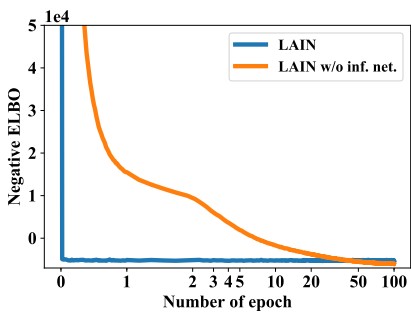

**Figure 3:** ELBO trajectories of LAIN with and without inference networks.

## 4.2 BIRD ABUNDANCE ESTIMATION

In this experiment, we estimate the spatial abundance of a bird species (*Savannah Sparrow*) using eBird dataset (Munson et al., 2015). The dataset has 14,393 observations, each of which is a reported bird count at a GPS location. We model the observed counts with GPS locations as the input. We set the likelihood to be a Poisson distribution, with rate given by $\lambda_i = \exp(f_i)$.

We compare different inference methods in terms of Negative predictive Log-Likelihood (NLL, the smaller the better predictive performance). Table 1 shows the results. We can see that LAIN achieves the best predictive performance at $K = 40$. Methods based on inducing points generally perform worse. In this dataset, observations have strong correlations in local areas, but inducing points are not efficient to capture the posterior at such a fine scale. In terms of running speed, LAIN is comparable to AIGP and DGP but faster than other methods. In our experiment, we have also tried to increase inducing points for DGP, but it does not improve its performance.

In this experiment, we also investigate whether inference networks work correctly. We run LAIN without inference networks and optimize $\mu$ and $L$ for the variational distribution directly. Then we compare LAIN models with and without inference networks by checking their optimization procedure. In this task, we fix hyperparameters, so the two methods solve a pure inference problem. Figure 3 is the trace plot of the negative ELBO versus training epochs. The figure shows the ELBO of the two LAIN models eventually converge to very similar values, though the ELBO without inference networks is slightly better after 50 epochs (likely due to the amortization gap). LAIN with inference networks significantly reduces the number of optimization epochs – the inference networks are well trained after only 0.01 epoch (about 100 iterations). In summary, the result indicates that inference networks can effectively identify the variational parameters using local information.

### 4.3 PRECIPITATION LEVEL ESTIMATION

In this task, we evaluate LAIN on a rainfall dataset. We process the precipitation dataset (Climate Data Online) and obtain the average precipitation level in May at 8,832 stations that are spatially distributed in the US. The GP inputs are GPS locations of these stations, and the observations are the average precipitation levels. We use the log-normal distribution as the likelihood, with its mean as function value $f$ from GP and variance as a hyperparameter learned from the data.

Table 2 summaries the experimental results. LAIN has better predictive performance, and its running speed is comparable to or faster than other methods.

We also analyze the goodness of our prior approximation since we can compute the exact $L_{cross}$ on this dataset. We compute $L_{cross}$ with the optimized $q(\mathbf{f})$ distribution as well as $\tilde{L}_{cross}$. The true value $L_{cross}$ and the approximation $\tilde{L}_{cross}$ are: 5,465 versus 5,396 when $K = 10$, 9,905 versus 8,490 when $K = 20$, and 10,086 versus 9,009 when $K = 40$. This result indicates that the approximation $\tilde{L}_{cross}$ is relatively accurate. Furthermore, $\tilde{L}_{cross}$ tends to be smaller than the true value and can be considered as a lower bound in such cases.

### 4.4 HAND-WRITTEN DIGIT CLASSIFICATION

In this experiment, we explore a high-dimensional inference problem, GP classification of MNIST digits (LeCun & Cortes, 2010). We consider a binary classification on handwritten images of 5 and 8. To make performance values of different methods more differentiable, we randomly choose a subset of size 7,858 from the original dataset. Pixel values are normalized to [0,1] in the preprocessing step. In the results, we also report the accuracy obtained by different methods.

The results are shown in Table 3. We see that LAIN performs the best in terms of classification accuracy. Its predictive NLL and running speed also overperform competing methods, though not very significant. We also observe that AIGP makes less confident predictions than other methods, which accounts for its worse predictive NLL but high accuracy. We do not report results from VFF due to memory issues.

We also examine KNN in this experiment. From the results, we notice that a small number of neighbors are often sufficient for KNN and LAIN models to perform well. By checking the running time of KNN, we also see that the time of finding nearest neighbors is only a small fraction of the total inference time on this dataset. There are slight differences regarding the test accuracy between KNN and LAIN, presumably due to different weighting schemes: LAIN weights different nearest neighbors according to their correlations, while KNN treats all nearest neighbors uniformly.

## 5 CONCLUSION

In this work, we propose a novel approach for GP inference. We construct a variational distribution that has a sparse decomposition on its covariance matrix. With this distribution, function value at a data point is inferred from its nearest neighbors, encouraging the inference efficiently focuses on approximating strong correlations posed by the prior. The proposed variational distribution is expressive to approximate the GP posterior and also provides a decent structure for efficient ELBO optimization. We further decompose the ELBO into homogeneous subtasks and therefore enable stochastic optimization. Finally, we devise inference networks to perform these subtasks and significantly reduce the number of variational parameters. Our proposed method performs well in terms of predictive performance and running speed on a series of benchmark tasks.

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
