# OpenReview forum: "Localizing and Amortizing: Efficient Inference for Gaussian Processes"
_ICLR.cc/2020/Conference — Reject_

### Official Review · AnonReviewer3 · 2019-10-22
**Official Blind Review #3**

**Rating:** 3

**Review:**

In this paper, the authors propose a family of variational distributions in which the variational covariance matrix is parameterized as RR', with R_ij being nonzero only when j is a neighbor of i as defined by prior covariance. In other words, data points are only allowed to have nonzero posterior covariance if they are highly correlated a priori. This results in a sparse factor of the covariance matrix which can be used for efficient computation. Rather than being parameterized directly, the variational parameters \mu_{i} and R_{i, n(i)} are parameterized using a GCN: the labels and prior covariance information are supplied to the GCN, which produces the mean for a data point x_i and the |n(i)| nonzero elements of the covariance factor R.

Overall, the idea is interesting in the sense that a variational family with enforced sparsity likely leads to a reasonable probabilistic model. However, I have a few concerns about the execution.

First, as a minor point, in my opinion, the approximations eqns. 3-7 are not sufficiently motivated, and are essential to the method as they allow for stochastic optimization. It would be useful to see an empirical analysis of the tightness of the additional approximations, as well as a generally expanded discussion in this section. Beyond this, the method is highly engineered but the only ablation study done of the various components is Figure 3, which merely offers an analysis of convergence speed, but not on final model performance. Both ideas introduced in the paper (localization and amortization) can be used independently of the other.

More importantly, I believe the experimental evaluation should be substantially broadened. At present, three datasets are considered and on one of them (MNIST) three methods considered are within the error bars of each other: the bolding in Table 3 is inappropriate. Many popularly used benchmark datasets for sparse GP methods are widely available, and it seems particularly essential to include datasets larger than the ones considered here, since exact GPs can trivially be trained on these datasets in a matter of seconds (at least for the regression tasks). Again ignoring the single classification task, Variational GP methods are usually only considered for regression for much larger datasets. I suspect that, with proper hardware (e.g. a GPU) and truly large datasets much of the speed advantage enjoyed by LAIN as reported in the paper will be lost to overhead.

**Experience Assessment:**

I have published in this field for several years.

**Review Assessment: Checking Correctness Of Derivations And Theory:**

I carefully checked the derivations and theory.

**Review Assessment: Checking Correctness Of Experiments:**

I carefully checked the experiments.

**Review Assessment: Thoroughness In Paper Reading:**

I read the paper thoroughly.

---

> ### Author Response · Authors · 2019-11-14
> **Re: Official Blind Review #3**
>
> Thank you for your valuable reviews. We would like to address your concerns as follows.
>
> 1. Motivation for Eq. 3-7:  Our main motivation stems from the nearest neighbors for GP inference. At the same time, we found it is also possible to leverage the nearest neighbors to enable stochastic optimization, which further motivates Eq. 3-7.
>
> 2. Goodness of approximation: In our experiment 4.3 (precipitation dataset), we showed the tightness of our prior approximation is relatively accurate. More interestingly, we found the approximate L_cross term tends to be smaller than the true value on that precipitation dataset.
>
> 3. Ablation tests: Figure 2 provided evidence of why localized inference works, and figure 3 evaluated the effectiveness of the inference networks. We will refine our manuscript with a clear discussion of the ablation tests.
>
> 4. Large dataset experiments: We agree with your statements and will make relative comparisons. But we expect our conclusion would not change so much. For the inducing-point methods, when there are a large number of data points but limited inducing points, the over-smooth problem we showed in the third plot of figure 2 becomes more severe. On the contrast, our method can still deal with such situations by considering nearest neighbors.

---

### Official Review · AnonReviewer1 · 2019-10-23
**Official Blind Review #1**

**Rating:** 1

**Review:**

PAPER SUMMARY:

This paper proposes a fast inference method for Gaussian processes (GPs) that imposes a sparse decomposition on the VI approximation of the posterior GP (for computational efficiency) using the KNN set of each data point. This is further coupled with armortized inference for better scalability.

NOVELTY & SIGNIFICANCE:

This paper adopts a different approach of characterizing the VI approximation of a GP posterior than original VI approximation that was developed in Titsias (2009): Instead of characterizing the surrogate q(f_I) of p(f_I | Y) for a small collection of inducing inputs, the proposed method characterize q(f) directly where q(f) = int_f_I q(f_I) p(f | f_I)df_I.

This is, however, a somewhat strange direction which, to me, seems to raise extra issues that could have been avoided if one follows the conventional VI approximation:

(1) As the posterior surrogate is now directly over f instead of f_I, the number of variational parameters is now proportional to the data size which requires several (redundant) extra approximations including armortized inference & the lower-bound on the entropy term that admits a sparse decomposition.

(2) This at least creates the armortized and entropy gaps that decrease the expressiveness of the original VI. While I understand that this is in exchange for the ability to encode local information (via KNN) within the surrogate posterior, it is not clear to me why do we need to incur all these computational issues to incorporate such local information.

(3) For example, instead of forcing such local information in the posterior surrogate q(f), we could alternatively let it be reflected in the test conditional p(f_* | f_I, Y_n(*)) such that the test output depends on both the inducing output and a local partition of data (e.g., via K-mean), which has been previously explored in [*] and later incorporated in the conventional VI paradigm of Titsias (2009) without incurring extra intractability [**].

(4) This maintains the dense correlation between data points within the same neighborhood while allowing the VI surrogate to be more concisely specified and independent of the no. of training data points. Furthermore, it also helps avoid the data-bound overhead of computing a KNN per test point.

[*] Local and global sparse Gaussian process approximations (AISTAT-07)
[**] A distributed variational inference framework for unifying parallel sparse gaussian process regression models (ICML-16)

To summarize, the practical significance of placing such a VI approximation directly on q(f) to encode such (indirectional) local information is, given the above, questionable to me.
Please note that I am not disputing the potential use of this VI form here, which could have been the only way to encode a different (directional) type of information.
For encoding KNN information, however, it only seems to create more troubles than it solves.

Minor point:

The above references, especially [*], should have been cited.

TECHNICAL SOUNDNESS:

[A] Optimization of the ELBO:

(1) The ordering of data (i.e., the directional information) was mentioned repeatedly in the paper but its importance to the fast approximation was neither explained nor discussed.
(2) The decomposition form of Eq. (6) also raises a question: How do we know that the term inside the log is positive? There seems to be missing information on the constraint of R.

[B] Amortized Inference:

(1) The choice of the GCN seems arbitrary here. I am in fact not sure why GCN is necessary for the inference network & furthermore, GCN also brings to the table another heuristic choice of A.
(2) How do we set the adjacency graph A?
(3) How do we know what is the right complexity for the GCN?

[C] Complexity:

The complexity analysis is too informal and lacking fine-grained information.
Please include a detailed complexity analysis of the training and inference cost in terms of the input dimension, the no. of data points, the size of the neighborhood and the batch size.
It is also necessary to factor in the KNN overhead (e.g., the cost of building the K-D tree for low-dimensional embedding of data & the approximation cost of projecting that information to high-dimensional data)

EXPERIMENT:

The experiment results only show marginal improvement over the baselines, and the size of the dataset for regression is too small. If I read correctly, both have fewer than 20000 data points.
SVGP in particular has been tested on a much larger datasets (AIRLINE, UK Housing) comprising millions of data points -- comparison on such dataset should have been reported.

Note that the largest dataset used to evaluate the efficiency of fast approximation of GP is on the scale of 6M data points [****]. On that note, eBird and precipitation should not even be considered mid-sized.

To demonstrate the efficiency of local information encoding, comparison with [*] should be reported. There is another class of inducing-point methods that use expectation propagation
that should have been discussed and/or compared with:

[***] A Unifying Framework for Gaussian Process Pseudo-Point Approximations using Power Expectation Propagation (JMLR-18)

[****] Distributed Gaussian Processes (ICML-15)

CLARITY:

The paper is clearly written.

REVIEW SUMMARY:

This paper adopts a VI approximation that deviates from the conventional form of (Titsias, 2009) to encode the KNN information, which causes extra computational issues (that incurs extra approximations). I find this deviation redundant seeing that the same information could have also been accounted for using the old VI form, which is a lot more computational efficient. I also find the experiment lacking as comparison with fast approximation method such as [*] that incorporate local information is not included. There are also a few technical ambiguities that need to be clarified.

------ Post-Rebuttal Update ------

Thank you for the rebuttal & I have read it in detail. However, it still does not address my concern, which I re-summarize here:

I do not dispute the beneficial of exploiting neighborhood information but my point is we could still leverage the same amount of neighborhood information without going through the trouble of incurring extra steps of approximation due to approximating q(f) instead of q(f_I) -- I think I have elaborated this in points (1) - (4) in my original review -- which also creates the amortized gap. I am also not sure what model reuse means (in this context) and whether it is relevant since it appears somewhat noncentric to the objective of this paper.

Also, apparently, the experiment is still lacking since to me, comparing with [*] is important in substantiating the contribution claim of this paper.

**Experience Assessment:**

I have published in this field for several years.

**Review Assessment: Checking Correctness Of Derivations And Theory:**

I carefully checked the derivations and theory.

**Review Assessment: Checking Correctness Of Experiments:**

I carefully checked the experiments.

**Review Assessment: Thoroughness In Paper Reading:**

I read the paper thoroughly.

---

> ### Author Response · Authors · 2019-11-14
> **Re: Official Blind Review #1**
>
> Thanks for your detailed feedbacks. We would like to address your concens as follows.
>
> 1. Novelty & Significance: We want to ease your concerns with two points. First, considering neighborhood information for GP inference is beneficial, as have been pointed out in the previous local GPs (e.g., [*]). In our work, we further show the importance of using the nearest neighbors for the inference of each data point (see the last three plots of Figure 2). Second, compared with previous local GPs that are using local partitions of data, our model optimizes a unified inference objective and also enables model reuse. Model reuse significantly increased the training speed.
>
> Although it looks like our inference objective has more approximations compared to the inducing-point methods, numerically we may have better results in practice (Figure 2 is one evidence).
>
>
> 2. Optimization of the ELBO:
> * The ordering of data: The last two plots of figure 2 validate the importance of data ordering. The proposed method (nearest neighbors based, data ordering free) is more stable compared to AIGP (data ordering dependent). Moreover, the better NLL-s achieved by our method than AIGP is another evidence of the significance of data ordering.
> * Term inside the log is positive in Eq. (6): the term inside the log is positive, because it is the variance of a conditional distribution q(\epsilon_i | f_n(i)). Note that the joint q(\epsilon_i, f_n(i)) is a multivariate Gaussian distribution.
>
> 3. Amortized Inference:
> * The choice of inference network: The inference network takes two inputs for a single data point: the observations at its neighbors and the kernel matrix defined on these neighbors. The two inputs form a weighted fully connected subgraph, for which GCN is an appropriate choice.
> * How do we set the adjacency graph A: We want A to capture the correlations between pairs of points. So we set A as the kernel matrix defined on a data point and its neighbors.
> * Right complexity for the GCN: We found our method is not sensitive to GCN structures in our experiment: A small GCN is able to perform well across all tasks.
>
> 4. Complexity analysis: Thanks for this feedback, we will refine the complexity analysis in details. But our general idea still holds for this analysis. First, one gradient computation of the ELBO takes constant time with stochastic optimization and constant size inference networks. Second, KNN often takes a much smaller fraction of time compared to the overall optimization procedure (see table 3 running time between LAIN and KNN). Note that we only need to run KNN once during the optimization.
>
> 5. Experiment: We agree with your comments, and we will make relative comparisons accordingly. But we expect our conclusion would not change so much. For the inducing-point methods, when there are a large number of data points but limited inducing points, the over-smooth problem we showed in the third plot of figure 2 is more severe. On the contrast, our method can still deal with such situations by considering nearest neighbors. For the local GPs like [*], our method still benefits by considering a unified inference objective with model reuse.
>
> [*] Local and global sparse Gaussian process approximations (AISTAT-07)

---

### Official Review · AnonReviewer2 · 2019-10-27
**Official Blind Review #2**

**Rating:** 3

**Review:**

1) Summary
The manuscript proposes a k-nearest-neighbor (KNN) Gaussian process (GP) approximate inference scheme to render computations more scalable.

2) Quality
Although the application is clear and the methodology is well established, the quality of the submission can be improved by a more thourough empirical evaluation in particular a proper evaluation in terms of runtime, approximation accuracy and comparison to baseline methods.

3) Clarity
The manuscript is reasonably well written and most of the technical and experimental content is accessible. There are some typos and some glitches in the notation. See "Details". There are some open issues regarding the KNN computations. See Questions.

4) Originality
The use of a localized (in the KNN sense) set of inducing inputs to improve GP inference but the impact needs to be better quantified empirically.

5) Significance
The proposed method is aiming at improving the established setting of GP inference. The modification is rather marginal and the empirical evaluation makes it hard to judge the relative merit of the proposal.

6) Reproducibility
The data is from published sources (toy, ebirds, precipitation, digits) and the code for the baseline methods and for the LAIM method itself is available. However, there is no code for the experiments, which makes the results slightly tricky to exactly reproduce.

7) Evaluation
The evaluation does not consider simple baselines like dense GPs or sparse approximations such as FITC and VFE. Also plain NN should be considered.

8) Questions
  A) How do you set the parameter delta?
  B) How are the nearest-neighbors computed in the first place? Does it require computing the dense covariance matrix?
  C) How accurate is the NN computation? How much of the computational effort (percentage) of the overall pipeline is required for the NN computation?
  D) How much better is the proposed approach than directly using NN predictions?
  E) Does "most correlated" in the footnote on page 2 really mean correlation or is it about covariance? The latter would involve a diagonal rescaling of the covariance matrix.

9) Details
  a) Abstract: "Gaussian Processes" -> "Gaussian processes"
  b) Intro: "GP poses a Gaussian prior" -> funny sentence, "with some special "
  c) Intro: "with some special structures" -> fix
  d) Background: "q(f)~N(mu,V)" -> imprecise notation, rather "q(f)=N(mu,V)"
  e) Background, footnote: "distance metrics" -> Are you talking about "distance" or "metric"?
  f) "Experiment" -> "Experiments"
  g) References: capitalization not correct e.g. Gaussian, Fourier, Bayes

**Experience Assessment:**

I have published in this field for several years.

**Review Assessment: Checking Correctness Of Derivations And Theory:**

I assessed the sensibility of the derivations and theory.

**Review Assessment: Checking Correctness Of Experiments:**

I assessed the sensibility of the experiments.

**Review Assessment: Thoroughness In Paper Reading:**

I read the paper thoroughly.

---

> ### Author Response · Authors · 2019-11-14
> **Re: Official Blind Review #2**
>
> We appreciate your insightful comments. We would like to address your concerns as follows.
>
> 1. Significance: To our best knowledge, the current work that combines localized neighbors and amortized inference for GP inference is Liu & Liu (2019). Compared to this work, we have two important improvements: 1) using K-nearest neighbors for the inference at each data point, and 2) eliminating the requirement of an ordering of data points.  These two seemingly simple conceptual improvements require substantial technical innovations: decomposable approximation of the prior with random orders and the HVI lower bound of the entropy, which are the technical contributions of this work.
>
> 2. Reproducibility: We shared the experiment file (experiment.py) in our submission. Running guidelines are also provided in this file.
>
> 3. Evaluation: We considered various GP inference methods in our experiments. Dense GP is examined on a toy dataset. Three recent inducing-point methods (SVGP, SAVIGP, and DGP) are included across all experiments. Fourier feature based approach (VFF) is also considered. We also evaluated plain NN on the MNIST dataset.
>
> 4. Delta value: We set delta to be a constant 1e-3 in our experiments. We add delta only to ensure a valid entropy lower bound, otherwise log q(f|\epsilon) in Eq. (5) will be deterministic.
>
> 5. Nearest-neighbors computation: In our experiments, we consider a euclidean distance based kernel (RBF). So we compute nearest neighbors either with k-d trees or with brute force, depending on the dimensionality of the dataset. There is no need to calculate the dense covariance matrix in our case.
>
> 6. NN accuracy and computation complexity: We compute the exact NN in our experiments. The first three datasets have at most two dimensions (GPS locations), so the computation efforts of NN are cheap using k-d trees. For the MNIST dataset, which has 784 (28 * 28) dimensions per image, the NN computation takes a much small fraction of time compared to the entire optimization procedure (see table 3 running time between LAIN and KNN).
>
> 7. Comparison with NN methods: We compared the proposed methods with KNN on the MNIST dataset. The results are shown in table 3. We found little difference when using a small number of neighbors (K=10). However, when the number of neighbors is relatively large (K=20 or K=40), some irrelevant neighbors may be included. Our methods improved performance by weighing different neighbors; KNN dropped its performance, perhaps the result of treating neighbors equally.
>
> 8. Most correlated: We mean the most correlated in the covariance.
>
> 9. Other details: Thanks for these comments. We have corrected these details in our manuscript.

---

> > ### Comment · AnonReviewer2 · 2019-11-15
> > **Nearest Neighbor Computations**
> >
> > Exact high-dimensional nearest neighbor computations are a hard in computer science, still.
> > For efficient inference, one needs a trade-off between 1) the closest neighbors in terms of covariance and 2) also a little more distant neighbors to not focus entirely on the environment of the point only e.g. when points are clustered and all neighbors carry essentially the same information which breaks the correlation structure.
> > Both aspects are missing in the NN-approximation approach presented here: The first aspect can only be computed for distance-based covariances of the type $k(x-y) = k( ||x-y|| )$ which is a serious limitation in practice. The second aspect is simply not present.
> > Implications are not discussed in the manuscript.

---

> > > ### Author Response · Authors · 2019-11-15
> > > **Re: Nearest Neighbor Computations**
> > >
> > > Thanks for your follow up. We believe these two points are valuable to our approach and will clarify our manuscript accordingly.
> > >
> > > 1. Although some extra efforts are needed to find neighbors for non-distance-based kernels, distance-based kernels (like RBF, Matern, rational quadratic, etc.) are widely used and is able to capture properties of various datasets. Our preliminary experimental results also suggest distance-based kernels perform reasonably well on four tasks, including a 1-D toy example (good predictive surfaces in figure 2), two spatial datasets, and MNIST (achieved >=98% classification accuracy). So we focus on the distance-based kernels in this work. In terms of the high dimensional data, existing approximate algorithms can be considered to speed up NN search [*, **].
> > >
> > > 2. We agree examining further neighbors might improve our model performance. For example, [***] considered the same prior approximation as ours, while they used a heuristic way for neighbor finding.  We can adapt similar ideas to improve our model flexibility, but our main conclusion will not change much. From another perspective, the situation like "when points are clustered and all neighbors carry essentially the same information" can be regarded as a sign of strong confidences within local areas. So prediction based on these local areas is also reasonable.
> > >
> > > [*] An optimal algorithm for approximate nearest neighbor searching fixed dimensions (JACM-1998)
> > > [**] Locality-sensitive hashing scheme based on p-stable distributions (Proceedings of the twentieth annual symposium on
> > > Computational geometry-2004)
> > > [***] Approximating likelihoods for large spatial data sets (Journal of the Royal Statistical Society: Series B Statistical Methodology-2004)

---

### Public Comment · ~Jiaxin_Shi1 · 2019-10-17
**Related work**

Hi,

Just want to point to our recent paper here
http://proceedings.mlr.press/v97/shi19a/shi19a.pdf
which is quite related to your work.

Best,
Jiaxin

---

> ### Author Response · Authors · 2019-10-21
> **Re: Related work**
>
> Thank you for pointing to your work here.  We agree that both your work and our submission use inference networks for GP inference. However, the underlying principles are very different.
>
> In our work, we want to keep the q distribution non-parametric. This property is important for cases where features do not provide much information. One example is our application of spatial data modeling: the input of two GPS coordinate often does not have enough information about labels. So our inference network explicitly uses labels of training instances as the input. By this construction, the complexity of our q distribution grows with the amount of data.
>
> Thank you again. We will cite your paper in the next version of our work.

---

### Decision · Program_Chairs · 2019-12-19

**Decision:**

Reject

**Comment:**

This paper presents a method for speeding up Gaussian process inference by leveraging locality information through k-nearest neighbours.

The key idea is well-motivated intuitively, however the way in which it is implemented seems to introduce new complications. One such issue is KNN overhead in high dimensions, but R1 outlines other potential issues too. Moreover, the method's merit is not demonstrated in a convincing way through the experiments. The authors have provided a rebuttal for those issues, but it does not seem to solve the concerns entirely.